# Evolutionary rate covariation is a reliable predictor of co-functional interactions but not necessarily physical interactions

Jordan Little[1], Maria Chikina[2], Nathan L Clark[1,3]*

[1]Department of Human Genetics, University of Utah, Salt Lake City, United States; [2]Department of Computational Biology, University of Pittsburgh, Pittsburgh, United States; [3]Department of Biological Sciences, University of Pittsburgh, Pittsburgh, United States

*For correspondence: nclark@pitt.edu

Competing interest: The authors declare that no competing interests exist.

**Abstract** Co-functional proteins tend to have rates of evolution that covary over time. This correlation between evolutionary rates can be measured over the branches of a phylogenetic tree through methods such as evolutionary rate covariation (ERC), and then used to construct gene networks by the identification of proteins with functional interactions. The cause of this correlation has been hypothesized to result from both compensatory coevolution at physical interfaces and nonphysical forces such as shared changes in selective pressure. This study explores whether coevolution due to compensatory mutations has a measurable effect on the ERC signal. We examined the difference in ERC signal between physically interacting protein domains within complexes compared to domains of the same proteins that do not physically interact. We found no generalizable relationship between physical interaction and high ERC, although a few complexes ranked physical interactions higher than nonphysical interactions. Therefore, we conclude that coevolution due to physical interaction is weak, but present in the signal captured by ERC, and we hypothesize that the stronger signal instead comes from selective pressures on the protein as a whole and maintenance of the general function.

## eLife assessment

This **useful** study seeks to address the importance of physical interaction between proteins in higher-order complexes for covariation of evolutionary rates at different sites in these interacting proteins. Following up on a previous analysis with a smaller dataset, the authors provide **compelling** evidence that the exact contribution of physical interactions, if any, remains difficult to quantify. The work will be of relevance to anyone interested in protein evolution.

## Introduction

The evolutionary rate of any protein-coding gene varies over time and hence between species. It has been observed that some genes have rates that covary over time with those of other genes and that they tend to be functionally related (*Clark et al., 2012*). Evolutionary rate covariation (ERC) is a measure of that correlation in relative evolutionary rates (RER) (*Kowalczyk et al., 2019*), which is the gene-specific evolutionary rate on a phylogenetic tree branch normalized by the genome-wide evolutionary rate on that branch. If one considers the set of branches relating the orthologous copies of a given gene, an ERC value measures how correlated its branch-specific rates of sequence divergence are with those of another gene.

Protein pairs that have high ERC values (i.e., high rate covariation over time) are often found to participate in shared cellular functions, such as in a metabolic pathway (*Findlay et al., 2014*; *Steenwyk et al., 2021*; *Steenwyk et al., 2022*) or meiosis (*Clark et al., 2013*) or being in a protein complex together. It was previously shown that physical interaction is not required for co-functional proteins to have correlated evolutionary rates, such as between the proteins of a metabolic pathway (*Clark et al., 2012*; *Juan et al., 2008*). However, it is still unclear whether the correlation between physically interacting proteins is strengthened by coevolution at their physical interface. Many co-functional proteins physically interact, such as those in protein complexes and enzyme–substrate interactions (*Gershoni et al., 2010*; *Fraser et al., 2004*; *Hakes et al., 2007*). For example, SMC5 and SMC6 form a complex and have a shared function in the spatial organization of chromatin. They also have similar variation in rates of evolution across the branches of a yeast phylogeny (*Figure 1A*). Their strong rate covariation is quantified as a Fisher transformed correlation coefficient (ftERC = 24.944). Given the statistical corrections performed, this value for SMC5-SMC6 is highly elevated because the expectation for noncorrelated pairs is zero. Yet, which forces led to this high correlation? Does their physical interaction play a role? Given the strong association between physical interactions and other indicators of co-functionality, the relative contribution of their physical interaction is difficult to dissect for SMC5 and SMC6, as well as for most physically interacting proteins.

There are two hypotheses addressing the relative contributions of physical interaction versus co-function to correlated evolutionary rates. The idea that physical interactions contribute more to correlated evolutionary rates hinges on the maintenance of proper binding (*Ramani and Marcotte, 2003*; *Salmanian et al., 2020*; *Goh et al., 2000*). Under this hypothesis, a mutation in one binding partner will result in a compensatory mutation in the other, that is, coevolution, consistent with the 'lock and key' model for maintenance of physical interactions (*Ramani and Marcotte, 2003*; *Goh et al., 2000*). Such coincident compensatory changes over long periods of time could lead to correlated changes in rate between the two physically interacting proteins. If the physical interaction hypothesis holds, then there is great interest in using rate covariation tools, such as ERC, to predict quaternary structure and connectivity in protein complexes. However, there is a competing hypothesis that diminishes that potential utility. The second hypothesis is that rate correlations are primarily the result of parallel changes in selective pressures acting upon all genes in that function. Shared selective pressures can result in correlated rates of evolution due to several underlying causes. These causes include relaxation of constraint on a function, which results in accelerated rates for the gene set providing that function. Similarly, increased importance of a function in a species could lead to increased constraint and result in a decrease in evolutionary rate in that species for all genes involved. Adaptive change to a specific function on a branch could also create a rate acceleration for that function's genes, specifically on that branch. Other sources of variation in selective pressure could result from changes in essentiality or expression level, network connectivity, among other forces.

For the purposes of this study, the forces that contribute to correlated evolutionary rates are grouped into two bins, physical and nonphysical. The physical force is coevolution occurring at physical interaction interfaces. Nonphysical forces include gene co-expression, codon adaptation, selective pressures, and gene essentiality. There is a well-accepted negative relationship between gene expression and rate of protein evolution where genes that are highly expressed generally have slower rates of evolution (*Drummond et al., 2006*; *Drummond et al., 2005*). However, *Cope et al., 2020* found that there is a weak relationship between both gene expression and the number of interactions a protein has with the coevolution of expression level. Conversely, they found a strong relationship between proteins that physically interact and the coevolution of gene expression. These findings illuminate the difference between the strong relationship of gene expression level on the average evolutionary rate of a protein and the weak contribution of gene expression level to correlated evolutionary rates of proteins across branches. The finding that physically interacting proteins have strong expression-level coevolution brings to question how much coevolution of physically interacting proteins contributes to overall covariation in protein evolutionary rates.

Previous studies have weighed whether physical coevolution is a strong contributor to rate covariation over time (*Hakes et al., 2007*; *Jothi et al., 2006*; *Kann et al., 2009*). The conclusions drawn from these studies are variable and, in some cases, contradictory. Furthermore, the studies had differing sample sizes and used different protein units to examine the physical interaction, such as surface residues (*Hakes et al., 2007*), the binding neighborhood (*Kann et al., 2009*), and protein domains (*Jothi*

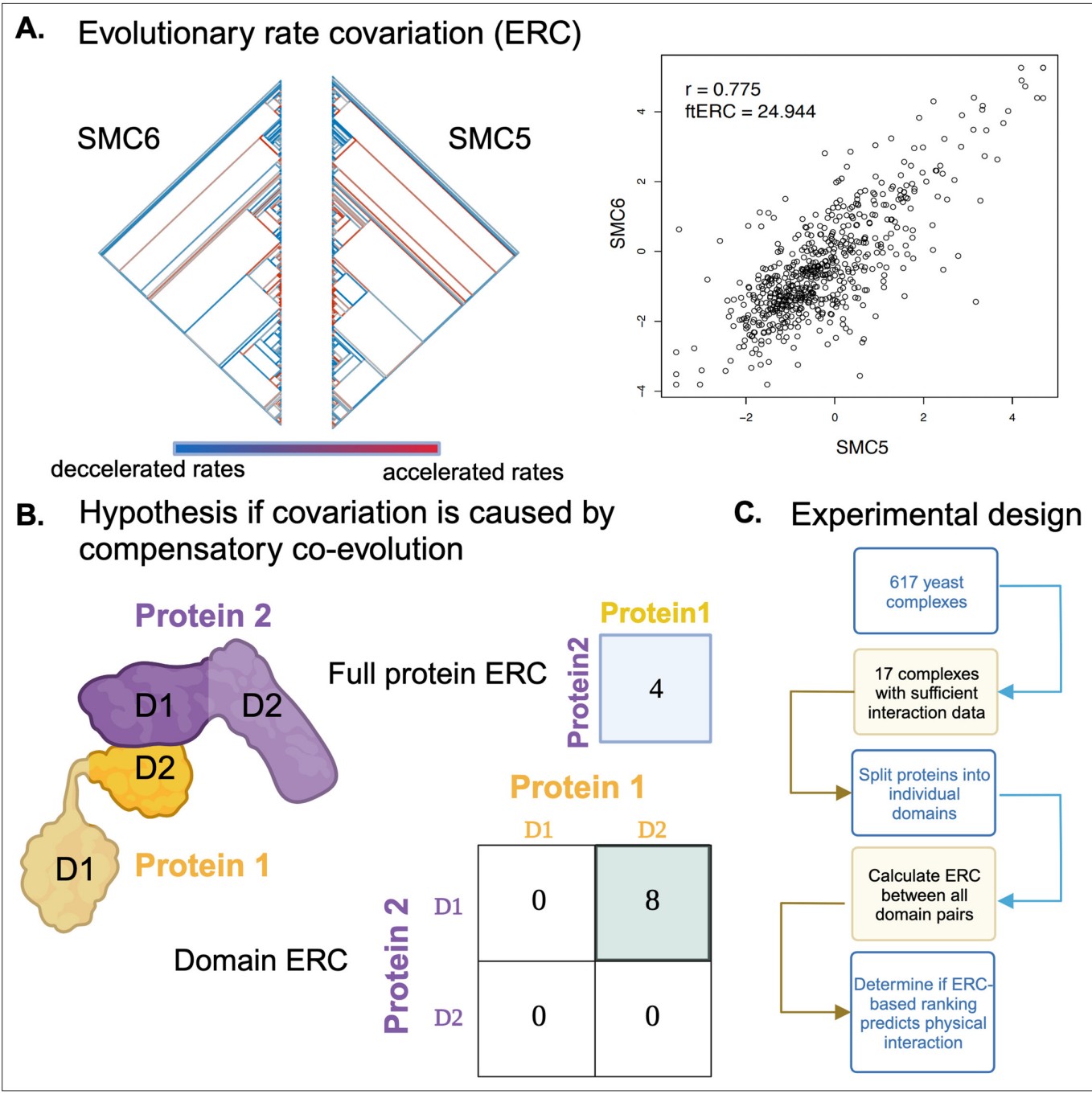

**Figure 1.** Overview of experimental schema and hypotheses. Proteins that share functional/physical relationships have similar relative rates of evolution across the phylogeny, as shown in (**A**) with SMC5 and SMC6. The color scale along the bottom indicates the relative evolutionary rate (RER) of the specific protein for that species compared to the genome-wide average. A higher (red) RER indicates that the protein is evolving at a faster rate than the genome average for that branch. Conversely, a lower (blue) RER indicates that protein is evolving at a slower rate than the genome average. The ERC (right) is a Pearson correlation of the RERs for each shared branch of the gene pair. (**B**) Suppose the correlation in RERs between two proteins is due to compensatory coevolution and physical interactions. In that case, the correlation of their rates (.ie., ERC value) would be higher for just the amino acids in the physically interacting domain. (**C**) Outline of experimental design. All panels were created with Biorender.com and published using a CC BY-NC-ND license with permission.

*et al., 2006*). It is difficult to compare the conclusions between these different experimental designs. The different units of the protein will, by nature, give different results. For instance, surface residues are under different constraints than buried residues, making it difficult to assess whether the physical interaction is driving the change or if it instead reflects selection on stability of the protein structure. Whereas examining protein domains gives a view of how the entire three-dimensional structure of the protein is potentially affected by changes in the binding partner. Given the contradictory conclusions and lack of statistical power in previous studies, the overall question regarding the contribution of coevolution to the overall rate covariation remains unanswered.

In this study, we test the contribution of compensatory physical coevolution to rate covariation by measuring ERC on a large dataset of 343 yeast species. Specifically, we ask whether physically interacting domains have higher ERC than domains of the same proteins that do not physically interact (*Figure 1B*). To illustrate the expectation being tested under the physical coevolution hypothesis, we present a hypothetical two-protein complex in *Figure 1B*. Only its two physically interacting domains would have an elevated ERC value above the null expectation of zero, or no correlation. Since domains can be analyzed separately, their ERC can be easily quantified in a practical workflow (*Figure 1C*). By looking only within complexes rather than across all proteins with annotated physical interactions, we normalize signals from functional associations, assuming that proteins in the same complex will be under the same functional constraints and, therefore, the same selective pressures and other nonphysical forces. Ultimately, we show a weak contribution from physical coevolution and, therefore, poor predictability of physical interactions based on ERC scores, regardless of complex size or average complex ERC. We further show that the ranking of physically interacting domains across the complex has little generalizability in predicting which domains or proteins physically interact.

## Results
### Protein pathways and complexes both have elevated ERC

Protein pathways and complexes are both functional units of the cell; however, complexes are defined by their physical interactions, while pathways may contain many proteins that do not physically interact at all. To investigate the discrepancy between contributions to ERC signal from co-function and physical interaction, we used a dataset of 343 evolutionarily distant yeast species. In total, 332 of the species are Saccharomycotina with 11 closely related outgroup species providing as much evolutionary divergence as between humans to roundworms (*Shen et al., 2018*). This dataset started with 12,552 orthologous genes, which we parsed into annotated pathways from KEGG (*Kanehisa et al., 2023*) and YeastPathway (*Cherry et al., 2012*) and protein complexes from the EMBL-EBI yeast complex portal (*Meldal et al., 2019*).

ERC was calculated for all pairs of the 12,552 genes. For each pair, the correlation is Fisher transformed to normalize for the number of shared branches that contribute to the correlation. This normalization is necessary to reduce false positives that have high correlation solely due to a small number of data points. This normalization also allows for direct comparison of ERC between gene pairs that have differing numbers of branches contributing to the score.

In general, ERC values for pathways and complexes were both high and had similar distributions after accounting for their sizes (*Figure 2A*). A majority of complexes (346 of 617) and pathways (109 of 125) had mean ERC values significantly greater ($p<0.05$) than a null distribution consisting of random size-matched protein sets (*Figure 2—figure supplement 1*). While protein complexes have higher mean ERC scores (median 5.366) than the pathways (median 4.597), the members of a given complex are also co-functional, making interpretation of the relative contribution of physical interactions to the average ERC score difficult.

To illustrate this difficulty at a more granular level, we present two examples, the SLIK protein complex and the motor protein pathway, both of which have significantly high average ERC values ($p<0.001$; *Figure 2B*). It is notable that when the ERC values for all physical interactions within the motor protein pathway are removed, it continues to have a significantly high average ERC. While the highest scoring protein pair in the SLIK complex, TAF5 and TAF6, physically interact and have a highly elevated ERC score (in top 1% genome-wide; *Figure 2—figure supplement 2*), the highest scoring pair in the motor protein pathway, TUB3 and TPM1, also have a highly elevated ERC score but do not physically interact (*Figure 2C*). This observation, which reflects the global pattern, makes it difficult to

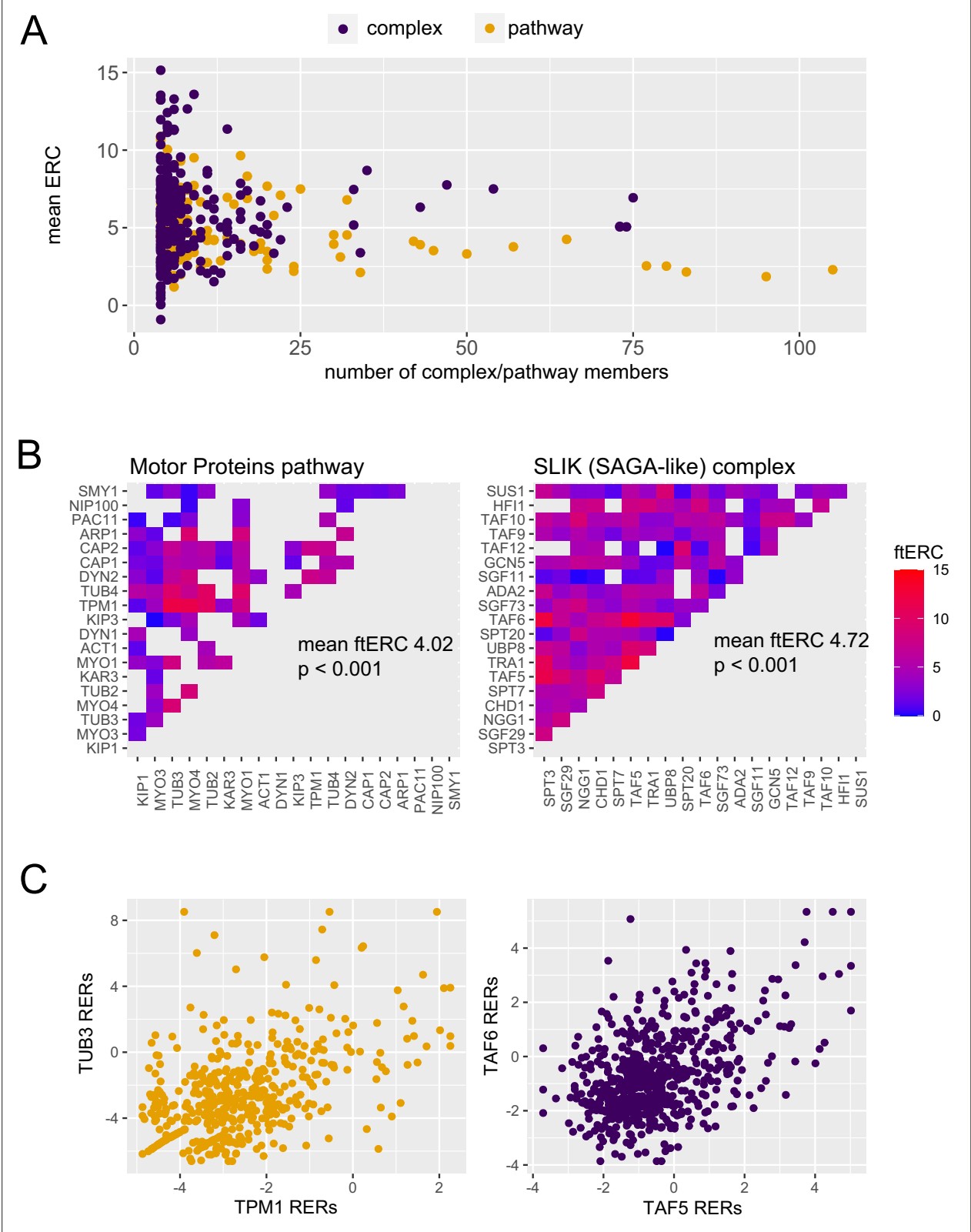

**Figure 2.** Protein complexes and cellular pathways have significantly high average evolutionary rate covariation (ERC). (**A**) The mean ERC values for 617 protein complexes (purple, median 5.366) and 125 cellular pathways (orange, median 4.597) versus the number of members contributing to the score. (**B**) Heat maps of the ERC scores for each protein pair in the motor proteins pathway (left) and SLIK complex (right). ERC for members of the motor

*Figure 2 continued on next page*

*Figure 2 continued*

proteins pathways that physically interact was set to NA (gray). (**C**) Scatter plots of the relative evolutionary rates for the top scoring pair from the motor proteins pathway (orange) and SLIK complex (purple).

The online version of this article includes the following figure supplement(s) for figure 2:

**Figure supplement 1.** Permutation p-value distribution of 617 protein complexes (**A**) and 125 pathways (**B**) when compared to a null distribution of 1000 samples.

**Figure supplement 2.** Histogram of the Fisher transformed evolutionary rate covariation (ERC) values for all 12,552 orthologous genes in the 343 yeast dataset.

determine whether the higher score between TAF5 and TAF6 is indicative of a stronger contribution from physical interaction to the ERC value or whether co-functionality is the main driver. These observations at both the global level and in individual complexes and pathways prompted further investigation into individual complexes to determine whether the physical interactions within a complex have higher average ERC scores.

## ERC does not distinguish physical interactions from nonphysical interactions within a given protein complex

To more directly test whether the signal contributing to the high rate covariation comes from the coevolution of physical interactions, we divided the proteins in a complex into their domains. A domain-level ERC analysis allowed us to directly contrast physically interacting domains with nonphysically interacting domains. To maintain the highest level of confidence in the analyses, we only selected complexes whose members had well-defined domain boundaries and annotated physical interactions between complex members ('Methods'). This resulted in a dataset of 14 complexes spanning functions from transcription and translation to autophagosome formation. We also added three complexes from *Jothi et al., 2006*: mitochondrial F1-ATPase, SEC23/24 heterodimer, and exportin CSE1 with substrate (*Table 1*).

We split the proteins from the 17 complexes into their annotated domains ('Methods') and calculated ERC between all domains in a complex. The average ERC for the physically interacting and nonphysically interacting domains for each complex was compared. Of the 17 complexes, 12 had higher average ERC for the physically interacting than the nonphysically interacting. When we look at an individual complex such as the COMA complex (*Figure 3*), the physically interacting domains have some of the highest ERC values. However, other physically interacting domains have some of the lowest ERC scores. To classify how often physically interacting domains have greater ERC and the statistical significance of such results, we proceeded with a rank-based analysis.

The rank-based method, receiver-operating characteristic (ROC) curve analysis, ranks the domain pairs based on their ERC score and calculates the true positive rate (TPR) and false positive rate (FPR) based on whether each pair is physically interacting (positive) or not (negative). The relationship between TPR and FPR is then plotted on a curve that scores the ability to rank a true positive above a false positive; the curve is summarized by the area under the curve (ROC-AUC). In our analysis, the ROC-AUC gives the proportion of times a random pair of physically interacting domains has a higher ERC score than a random pair of nonphysically interacting domains where a value of 1 would indicate that all physical interactions ranked above all nonphysical interactions.

Of the 17 complexes, 12 had an ROC-AUC > 0.5. Since the random expectation would be that half would exceed 0.5, this is a significant excess (binomial test, p=0.0245). Moreover, five complexes had AUCs > 0.7 (*Figure 4*). These results indicate that physically interacting domains tend to have a higher ERC than nonphysically interacting domains in those complexes. Likewise, four complexes had their physically interacting domains ranked significantly higher than noninteracting domains at an alpha of 5% (one-tailed Mann–Whitney *U* test), which is also a significant excess of complexes overall (binomial test, p=0.0012). Once again, this indicates a significant amount of ERC signal coming from physically interacting domain pairs within these complexes.

We then applied a third test for a general pattern among all 17 complexes. We generated a null ROC-AUC distribution by permuting the location of the positive class along the ranked list for each complex ('Methods'). We compared the true average ROC-AUC from the 17 complexes (0.596) to the permuted distribution of average ROC-AUCs, which resulted in a permutation p-value of 0.08. The

**Table 1.** Evolutionary rate covariation for 17 protein complexes and their physically interacting domains.

| Complex | Complex ERC | Complex permutation p-value | Physically interacting domain ERC | Nonphysically interacting domain ERC | Number of proteins (domains) |
|---|---|---|---|---|---|
| Eukaryotic translation initiation factor 3 core complex (*Politis et al., 2015*) | 11.539 | <0.001 | 8.852 | 11.125 | 6 (14) |
| MCM complex (*Frigola et al., 2017*) | 12.116 | <0.001 | 10.283 | 8.000 | 6 (26) |
| NUP84 (*Shi et al., 2014*) | 3.590 | 0.001 | 1.541 | 4.411 | 5 (9) |
| Origin of replication complex (*Feng et al., 2021*) | 5.420 | 0.012 | 5.250 | 4.035 | 6 (15) |
| PAN1 actin cytoskeleton-regulatory complex (*Complex Portal, 2023*) | 6.140 | 0.014 | 4.963 | 4.716 | 3 (13) |
| SMC5-6 SUMO ligase complex (*Yu et al., 2021*) | 8.813 | 0.004 | 5.153 | 4.110 | 4 (15) |
| TREX (*Xie et al., 2021*) | 5.368 | 0.02 | 4.157 | 4.129 | 5 (12) |
| EXOCYST (*Ganesan et al., 2020*) | 12.040 | <0.001 | 6.221 | 5.974 | 8 (21) |
| COMA complex (*Fischböck-Halwachs et al., 2019*) | 2.883 | 0.157 | 3.051 | 2.869 | 4 (11) |
| SWI/SNF chromatin remodeling complex (*Han et al., 2020*; *Schubert et al., 2013*) | 1.853 | 0.291 | 4.121 | 1.783 | 7 (20) |
| CUL8-MMS1-MMS22-CTF4 E3 ubiquitin ligase complex (*Mimura et al., 2010*) | 0.9925 | 0.175 | 2.611 | 1.576 | 5 (14) |
| GET4-GET5 transmembrane domain recognition complex (*Chang et al., 2010*) | 2.501 | 0.17 | 4.569 | 3.302 | 4 (7) |
| ATG17-ATG31-ATG29 complex (*Ragusa et al., 2012*) | 1.825 | 0.29 | 3.444 | 1.584 | 5 (9) |
| ESCRT-I complex (*Kostelansky et al., 2007*) | 2.752 | 0.137 | 4.041 | 2.402 | 4 (8) |
| Mitochondrial F1 ATPase (*Jothi et al., 2006*) | 11.015 | <0.001 | 4.883 | 5.654 | 3 (7) |
| SEC23/24 heterodimer (*Jothi et al., 2006*) | 9.321 | 0.004 | 10.530 | 7.754 | 2 (10) |
| Exportin CSE1 with cargo (*Jothi et al., 2006*) | 10.902 | <0.001 | 8.871 | 6.326 | 2 (12) |

ERC, evolutionary rate covariation.

lack of significance in the global permutation test suggests that the global signal for the contribution of physical coevolution to ERC is weak enough that it can be masked by the majority of complexes that show no evidence in favor.

## The power of ERC to detect physically interacting domains is not generalizable in single protein pairs

The wide range in ROC-AUCs from the previous analysis suggested the possibility of confounding factors masking the signal. Potential confounders could be members of the complex moonlighting in other pathways (*Mani et al., 2015*) or core membership versus peripheral membership within the complex itself (*Chakraborty et al., 2010*). To test for the effect of physical interaction free from these confounders, we broke each complex down and only compared the domains within two proteins at a time. Thus, the variation introduced by these potential confounders will be consistent across all domains between a single protein pair, and comparing only their domains pair-by-pair allows cleaner testing of the contribution of physical coevolution to the ERC signal. This analysis also allowed us to

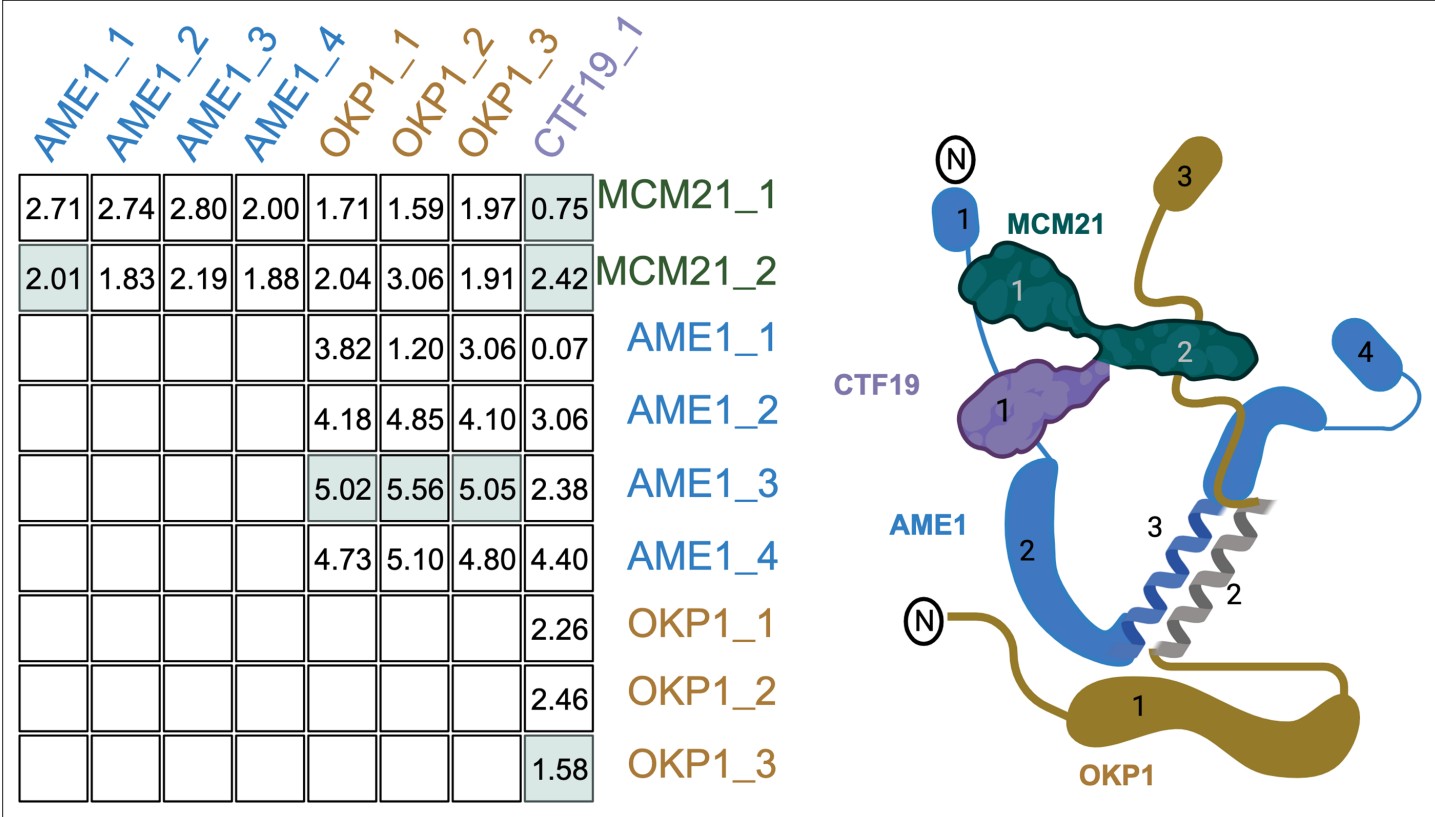

**Figure 3.** Recreation of *Figure 1B* with the COMA complex. The table shows the evolutionary rate covariation (ERC) values for each domain pair as labeled in the cartoon on the right. The domain pairs with physical interactions are highlighted in green.

determine the significance of a complex having a physical interaction ranked first. If the ranking is significant, it would indicate a future use case of ERC to predict physical interactions.

To determine the significance of the physically interacting domain ranking within each pair of complex proteins, we calculated the proportional rank of the physically interacting domain pair versus all other domain pairs ('Methods'). This metric gave us the proportion of times the physical interaction ranked higher than nonphysical interactions. The complexes had individual protein pair proportional rank values spanning the entire range of 0–1 without clustering at either extreme (*Figure 5*). These results indicate that even within the same complex, there is a wide variation in how strongly the physical interaction correlates with high ERC.

We then looked at the complex average proportional rank to determine whether there was some overarching signal. We took the average for all protein pairs within a complex to get the mean complex proportional rank. The significance of the complex proportional rank was determined by generating a null distribution of proportional rank values for each protein pair and randomly sampling from that to generate a complex-wide null distribution. The observed average for each complex was compared to the null. At a permutation p-value of 0.01, only two complexes had significantly high proportional rank, indicating a strong contribution from physical interactions: CUL8-MMS1-MMS22-CTF4 E3 ubiquitin ligase and exportin complexes (*Figure 5*). Three additional complexes were significant at a permutation p-value of 0.05: MCM, ORC, and ESCRT-I. This leaves over two-thirds of the complexes with no significant contribution from physical interactions to the ERC signal.

Given that some of the complexes ranked the physically interacting domains significantly higher in the proportional rank test, it suggests that compensatory co-evolution does contribute to the ERC signal. However, the inconsistency of the ranking indicates that there is not a consistent enough signal to confidently call an interaction physical or not and would be of little value to an experimentalist wanting to infer interacting domains. Ultimately, the contribution from physical interactions on the ERC signal is not strong enough to determine whether a high-ranking protein pair is associated due to physical interaction instead of nonphysical forces.

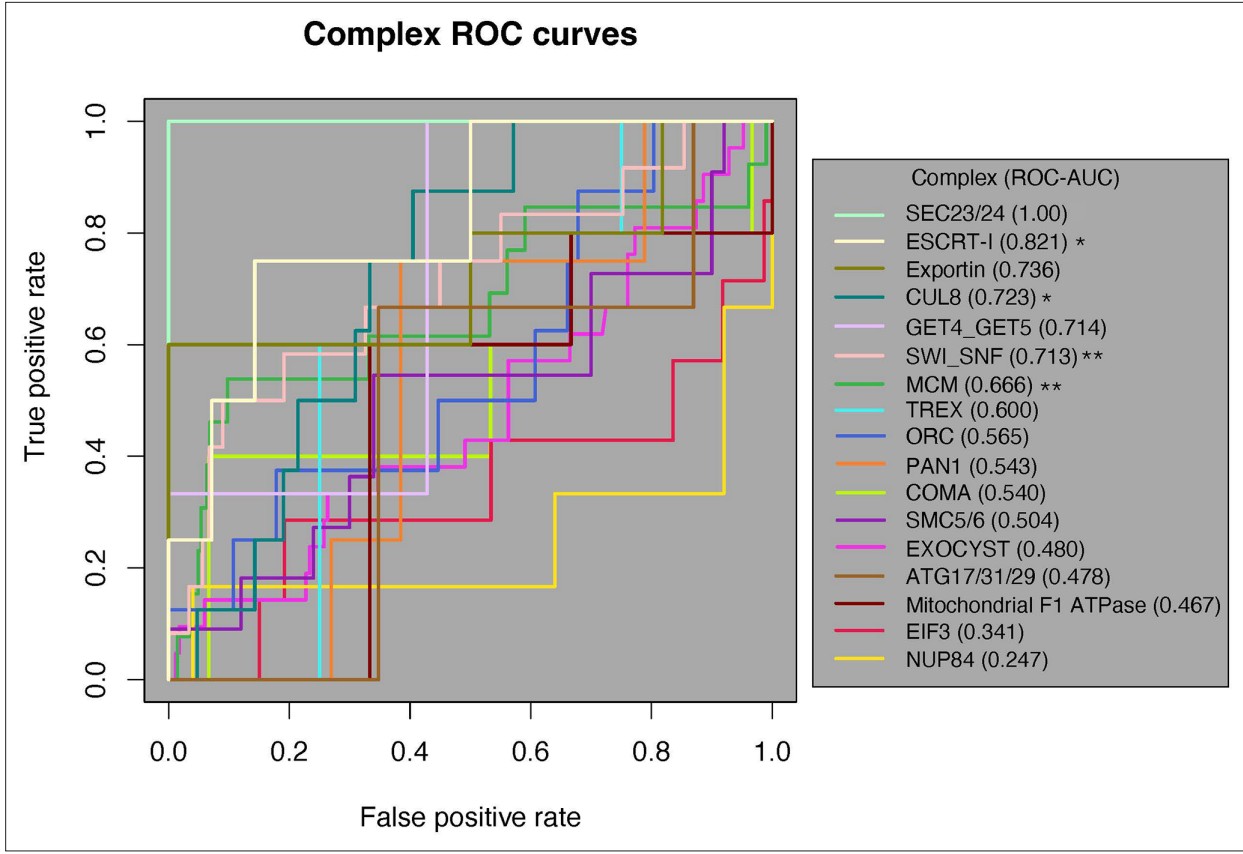

**Figure 4.** Receiver-operating characteristic (ROC) curve analysis of all 17 protein complexes. Of the 17 complexes, 12 have an ROC-AUC > 0.5. The SEC23/24 complex (bright green) has the highest ROC-AUC at 1, and the NUP84 complex (marigold) has the lowest AUC of 0.247. One-tailed Mann–Whitney *U* test, *p<0.05, **p<0.01. AUC, area under the curve.

## Discussion

Given the differing conclusions reached in previous studies about the strength of contribution from physical interaction to correlated evolutionary rates (*Hakes et al., 2007*; *Jothi et al., 2006*; *Kann et al., 2009*), we aimed to provide a robust conclusion with improved experimental design and sample size. Upon the addition of hundreds of species to the analysis, we were able to look at evidence of compensatory coevolution with increased power at a number of scales: whole protein, domains within complexes, and domains between individual protein pairs. We propose that using entire domains as a unit of study, rather than amino acids at physical interfaces, captures structural changes that could still impact the binding and would not be captured by just examining the binding residues themselves. These changes would still be selected for through compensatory coevolution, and relegating them to the noninteracting category could potentially mask correlated signals.

We found that compensatory coevolution due to physical interaction contributes to ERC but only in some complexes. Looking across all complexes in our study, a proportion higher than random chance showed elevated ERC between interacting domains; this result was reflected in 12 of the 17 complexes with ROC-AUCs over 0.5 and the 4 complexes with individually significant rankings. Moreover, when we examined only single protein pairs, we found a similar excess of high-scoring physically interacting domains. This indicates that there is a non-negligible signal from physical interaction contributing to ERC. Evidence for physical coevolution however was tempered by a global permutation test, which did not reach significance, indicating that this inference is sensitive to approach and further underlines the weak contribution of physical coevolution. In light of this weak contribution and inconsistency between individual protein complexes, the practical application of a tool such as ERC to provide actionable hypotheses for physically interacting domains is not advisable, given the uncertainty as to which complexes would have high ERC between physically interacting domains.

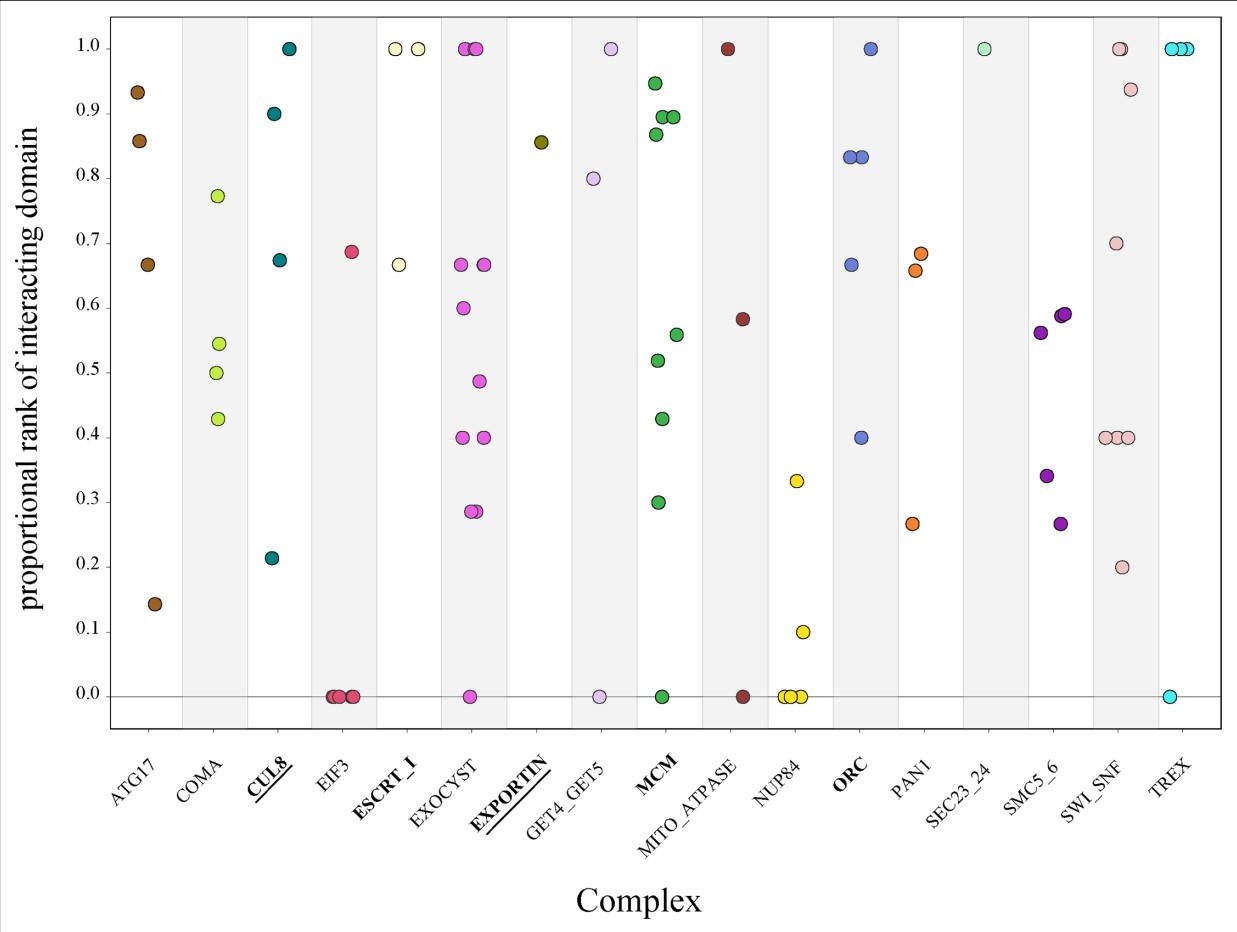

**Figure 5.** Protein-vs-protein physically interacting domains do not consistently rank higher than nonphysically interacting domains. The domains from individual protein pairs within each complex were ranked, and the proportional ranking of the physically interacting domain was calculated. Each dot represents the proportional rank of the interacting domain pair for a protein pair, with colors representing the compledataxes. A score of 1 indicates that the physically interacting domains were ranked first. A score of 0 indicates that the physically interacting domains were ranked last. Permutation test, p<0.05 (bold), p<0.01 (bold and underlined).

Previous studies attributed varying degrees of ERC signal to physical interactions between proteins. On one extreme, *Hakes et al., 2007* found no evidence that the physical interaction interface between two proteins had a greater correlation of evolutionary rates than the whole protein or just surface residues. They concluded this after examining surface and potentially interacting residues across 32 complexes in at least 12 eukaryotic species for each complex using the *Saccharomyces cerevisiae* sequence as the BLAST query. On the other extreme, *Jothi et al., 2006* concluded that there was a strong enough signal to predict which domains of a protein complex interact with significant accuracy; however, their analysis was limited to just three protein complexes with orthologs in at least 10 of 93 eukaryotic genomes. Because of their conclusions, we included their complexes in our own study and achieved similar rankings of the physically interacting domains. However, only one of their complexes was statistically significant in our analysis that we attribute to a lack of power given that one of the complexes only had one physically interacting domain pair out of 24 pairs. *Kann et al., 2009* reached a similar conclusion to our study using 70 bacterial species. They found that binding neighborhoods from 26 interacting pairs have a higher ERC on average than 1291 random nonbinding sequences of the same length, ultimately concluding that while physical interactions are not the sole contributor to evolutionary covariation, they still contribute strongly enough to be detected.

These previous studies agreed that compensatory coevolution is not the sole force behind ERC but they came to differing conclusions as to how much it contributes. This study concludes that there is a detectable but weak contribution from physical interactions that is not strong enough to confidently

predict which domains physically interact. This study also found that the contribution of physical interaction was limited to a minority of complexes despite the high power lent from using hundreds of species to calculate ERC; this inconsistent effect across complexes might explain why previous studies and this one found varying degrees of contribution since results could have depended on which complexes were chosen and which species were used. Among the complexes we found to have a physical coevolution component, there is no obvious reason why these specific complexes would exhibit stronger coevolution between physically interacting domains when compared to other complexes. We hypothesize that while compensatory coevolution is a known phenomenon (*Gershoni et al., 2010*; *Juan et al., 2008*), amino acid changes resulting from it are relatively rare and are unlikely to contribute greatly to the general ERC signal between co-functional proteins. In contrast, previous work finds evidence that relaxation of selective constraint can lead to drastic rate variation and hence a strong covariation signal (ERC) (*Clark et al., 2013*). Overall, variations in selective constraint, and other nonphysical forces, such as essentiality, expression level, codon adaptation, and network connectivity, seem to be the primary contributors to ERC. Indeed, the results of these forces are visible in the elevated ERC values of genetic pathways that do not physically interact (*Figure 2*).

## Methods

**Key resources table**

| Reagent type (species) or resource | Designation | Source or reference | Identifiers | Additional information |
|---|---|---|---|---|
| Software, algorithm | ERC | https://github.com/nclark-lab/erc/; copy archived at **Clark and Little, 2023** | | See 'Calculating ERC' |
| Other | 343 yeast amino acid sequences and trees | https://doi.org/10.1016/j.cell.2018.10.023 | https://doi.org/10.6084/m9.figshare.5854692.v1 | |

### Calculating ERC

ERC is calculated by correlating RERs between two gene trees using a Pearson correlation. The RER is the rate at which a branch on a gene tree changes compared to the genome-wide average and is calculated as described by *Kowalczyk et al., 2019*. The method limits comparisons to gene trees that share at least 15 species and requires that all trees have the same topology. Prior to correlation, the RERs are Winsorized, taking the three most extreme values and condensing them to the fourth. This reduces false positives that are due to single outliers skewing the correlation. After calculating the correlation between RERs for each gene pair, the correlation values are Fisher transformed using the equation

$$ftERC = arctan\left(correlation\right) * \sqrt{number\ of\ branches\ -\ 3}$$

This allows for a direct comparison between gene pairs that do not have the same number of branches contributing.

The full ERC pipeline can be found at https://github.com/nclark-lab/erc/tree/main/physical_interaction_paper.

### Complex and pathway ERC permutation test

We took the entire yeast complexome from the EMBL complex portal (*Meldal et al., 2019*) and curated yeast pathways from KEGG (*Kanehisa et al., 2023*) and YeastPathway (*Cherry et al., 2012*). These lists were then compared to our dataset and pared down to complexes/pathways that had greater than four members. This resulted in 617 protein complexes and 125 pathways.

We then ran 1000 permutations to find the significance of the ERC scores between complex/pathway members. The null distribution was generated by sampling the same number of random genes in the complex/pathway from the entire dataset. The average ERC from the complex/pathway was then compared to the null distribution to get a p-value.

### Preparing the protein complexes

Fourteen complexes were chosen by searching for protein complexes within the EMBL yeast complex portal with crosslink, crosslink/mass spectrometry, or crystallographic data. Three additional complexes

were added based on *Jothi et al., 2006*: mitochondrial F1 ATPase, SEC23/24 heterodimer, and the exportin CSE1P complexed with cargo. The physical interactions were collected from the literature sources for each complex .

Each protein was subdivided into domains by running the amino acid sequence through interpro scan (*Jones et al., 2014*). New gene trees were generated for each domain using phangorn (*Schliep, 2011*), and ERC was run to calculate an all-domain-by-all-domain matrix. We generated domain-vs-domain ERC matrices for each of the 17 complexes that were used for all analyses.

### Generating ROC curves

ROC curve analysis was performed using the PRROC package (*Grau et al., 2015*). First, the ERC matrices for each complex were turned into pairwise edge lists with columns [GENEA, GENEB, ERC, class] and ranked by ERC value. The positive class was defined as the physical interactions found in the primary reference for each complex and denoted with a '1'. The negative class was any protein domain pair without annotated physical interactions, denoted with a '0'.

The statistical significance of the ROC-AUC was determined by a one-tailed Mann–Whitney $U$ test, using the relationship between AUC and U defined by

$$AUC = \frac{U}{n_0 * n_1}$$

where $n_0$ is the number of nonphysically interacting domains and $n_1$ is the number of physically interacting domains.

ROC-AUC permutations were calculated by randomly shuffling the order of physical interactions within each complex ranked list 1000 times. The permuted AUC was calculated using the same pipeline as described above. The full study analysis was performed by taking the average from each of the 1000 permutations across all complexes.

### Calculating proportional rank of physically interacting domain pairs versus all other domain pairs

To test how often ERC ranks the physical interactions higher than nonphysical, we compared each pair of proteins individually, where each protein's domains were only compared to one other protein's domains if they shared a physical interaction somewhere along the full protein. For example, in the COMA complex, the domains for OKP1 were only compared to the domains of AME1 because OKP1 does not share a physical interaction with either CTF19 or MCM21. The matrix was then ranked by ERC score. The proportional ranking of the physically interacting domain pair was calculated by:

$$1 - \frac{1 - (rank\ of\ physical\ interaction)}{1 - (total\ number\ of\ pairs)}$$

A score of 1 indicates the physically interacting domain pair is ranked first. A score of 0 indicates the domain pair with a physical interaction was ranked last. If the protein pair had more than one domain pair that had a physical interaction, the average proportional rank score was taken. The observed complex proportional rank score is the average of each protein's score .

The permutation p-value for the complex proportional rank score was calculated by randomly shuffling the ranking of the physical interaction(s) between two proteins 1000 times and calculating the proportional rank each time to generate a null distribution. The null distribution for the entire complex was calculated by randomly selecting a proportional rank value from the null distribution of each protein pair from the complex and averaging it. The observed complex average proportional rank was then compared to this null distribution to calculate the p-value.

## Acknowledgements

This project was funded by the National Human Genome Research Institute at the National Institutes of Health (HG009299 to NC and MC). The support and resources from the Center for High Performance Computing at the University of Utah are gratefully acknowledged.

# Additional information

## Funding

| Funder | Grant reference number | Author |
|---|---|---|
| National Institutes of Health | HG009299 | Maria Chikina Nathan L Clark |

The funders had no role in study design, data collection and interpretation, or the decision to submit the work for publication.

## Author contributions
Jordan Little, Conceptualization, Data curation, Formal analysis, Validation, Investigation, Visualization, Methodology, Writing – original draft, Writing – review and editing; Maria Chikina, Software, Funding acquisition, Methodology; Nathan L Clark, Conceptualization, Supervision, Funding acquisition, Investigation, Writing – original draft, Project administration, Writing – review and editing

## Author ORCIDs
Jordan Little ⬚ http://orcid.org/0000-0002-8590-9357
Nathan L Clark ⬚ https://orcid.org/0000-0003-0006-8374

Reviewer #1 (Public Review): https://doi.org/10.7554/eLife.93333.3.sa1
Reviewer #2 (Public Review): https://doi.org/10.7554/eLife.93333.3.sa2
Author Response https://doi.org/10.7554/eLife.93333.3.sa3

---

# Additional files

## Supplementary files
• MDAR checklist

## Data availability
All data generated have been deposited on github at https://github.com/nclark-lab/erc (copy archived at *Clark and Little, 2023*).

---

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
