## [Editor Report · eLife assessment]

This **useful** study seeks to address the importance of physical interaction between proteins in higher-order complexes for covariation of evolutionary rates at different sites in these interacting proteins. Following up on a previous analysis with a smaller dataset, the authors provide **compelling** evidence that the exact contribution of physical interactions, if any, remains difficult to quantify. The work will be of relevance to anyone interested in protein evolution.

---

## [Referee Report · Reviewer #1 (Public Review)]

Summary:

In the manuscript titled "Coevolution due to physical interactions is not a major driving force behind evolutionary rate covariation" by Little et al., explores the potential contribution of physical interaction between correlated evolutionary rates among gene pairs. They find that physical interaction is not the main driving of evolutionary rate covariation (ECR). This finding is similar to a previous report by Clark et al. (2012), Genome Research, wherein the authors stated that "direct physical interaction is not required to produce ERC." The previous study used 18 Saccharomycotina yeast species, whereas the present study used 332 Saccharomycotina yeast species and 11 outgroup taxa. As a result, the present study is better positioned to evaluate the interplay between physical interaction and ECR more robustly.

Strengths & Weaknesses:

Various analyses nicely support the authors' claims.

---

## [Referee Report · Reviewer #2 (Public Review)]

Summary:

The authors address an important outstanding question: what forces are the primary drivers of evolutionary rate covariation? Exploration of this topic is important because it is currently difficult to interpret the functional/mechanistic implications of evolutionary covariation. These analyses also speak to the predictive power (and limits) of evolutionary rate covariation. This study reinforces the existing paradigm that covariation is driven by a varied/mixed set of interaction-types that all fall under the umbrella explanation of 'co-functional interactions'.

Strengths:

Very smart experimental design that leverages individual protein domains for increased resolution.

Weaknesses:

Nuanced and sometimes inconclusive results that are difficult to capture in a short title/abstract statement.

EDIT: The authors have done a satisfactory job of honing their language to get the nuanced ideas across clearly. The added scholarship and theoretical discussion they added strengthen the impact of the manuscript. The revised edition addresses my concerns.

---

## [Author Response]

The following is the authors’ response to the original reviews.

**Public Reviews:**

**Reviewer #1 (Public Review):**
Summary:The manuscript titled "Coevolution due to physical interactions is not a major driving force behind evolutionary rate covariation" by Little et al., explores the potential contribution of physical interaction between correlated evolutionary rates among gene pairs. The authors find that physical interaction is not the main driving of evolutionary rate covariation (ECR). This finding is similar to a previous report by Clark et al. (2012), Genome Research, wherein the authors stated that "direct physical interaction is not required to produce ERC." The previous study used 18 Saccharomycotina yeast species, whereas the present study used 332 Saccharomycotina yeast species and 11 outgroup taxa. As a result, the present study is better positioned to evaluate the interplay between physical interaction and ECR more robustly.Strengths & Weaknesses:Various analyses nicely support the authors' claims. Accordingly, I have only one significant comment and several minor comments that focus on wordsmithing - e.g., clarifying the interpretation of statistical results and requesting additional citations to support claims in the introduction.

We are pleased the reviewer found the analyses to support the claims. We have addressed comments related to clarifying interpretations as suggested in the Recommendations to the Authors. For example, we have added discussion and clarification on the other parameters that could affect the strength of ERC correlations.

Reviewer #2 (Public Review):

Summary:The authors address an important outstanding question: what forces are the primary drivers of evolutionary rate covariation? Exploration of this topic is important because it is currently difficult to interpret the functional/mechanistic implications of evolutionary covariation. These analyses also speak to the predictive power (and limits) of evolutionary rate covariation. This study reinforces the existing paradigm that covariation is driven by a varied/mixed set of interaction types that all fall under the umbrella explanation of 'co-functional interactions'.Strengths:Very smart experimental design that leverages individual protein domains for increased resolution.Weaknesses:Nuanced and sometimes inconclusive results that are difficult to capture in a short title/abstract statement.

We appreciate the reviewer’s acknowledgement of the experimental design. We have addressed the nuance of the results by changing the title and clarifying other statements throughout the manuscript as suggested in the reviewer’s recommendations. We have also addressed reviewer comments asking for further explanation on using Fisher transformations when normalizing the Pearson correlations for branch counts.

**Reviewer #3 (Public Review):**
Summary:The paper makes a convincing argument that physical interactions of proteins do not cause substantial evolutionary co-variation.Strengths:The presented analyses are reasonable and look correct and the conclusions make sense.Weaknesses:The overall problem of the analysis is that nobody who has followed the literature on evolutionary rate variation over the last 20 years would think that physical interactions are a major cause of evolutionary rate variation. First, there have been probably hundreds of studies showing that gene expression level is the primary driver of evolutionary rate variation (see, for example, [1]). The present study doesn't mention this once. People can argue the causes or the strength of the effect, but entirely ignoring this body of literature is a serious lack of scholarship. Second, interacting proteins will likely be co-expressed, so the obvious null hypothesis would be to ask whether their observed rates are higher or lower than expected given their respective gene expression levels. Third, protein-protein interfaces exert a relatively weak selection pressure so I wouldn't expect them to play much role in the overall evolutionary rate of a protein.

We thank the reviewer for their comments and suggestions. A point to immediately clarify is that the methods studied in this manuscript deal with rate variation of individual proteins over time, and if that variation correlates with that of another protein.. The numerous studies the reviewer refers to deal with explaining the differences in average rate between proteins. These are different sources of variation. It has not, to our knowledge, been shown that variation in the expression level of a single protein over time is responsible for its variation in evolutionary rate over time, let alone to a degree that allows its variation to correlate with that of a functionally related protein. That question interests us, but it is not the focus of this study.

In our study, we sought to test for a contribution of physical interaction to the correlation of evolutionary rate changes as they vary over time, i.e. between branches. We made many changes to clarify this distinction in our revisions.

We agree that the manuscript would be more clear to define the forces proposed to lead to difference in rate in general, which includes expression levels. We had generally considered expression level as one of the many potential non-physical forces, but failed to make that explicit and instead focused on selection pressure. In our revision we describe expression level as another potential driver of evolutionary rate variation over time. References to previous literature have been made in the introduction. We also added a more explicit explanation of the rate covariation over time that we are measuring in contrast with the association between expression level and rate differences between proteins that was studied in previous literature.

On point 3, the authors seem confused though, as they claim a co-evolving interface would evolve *faster* than the rest of the protein (Figure 1, caption). Instead, the observation is they evolve slower (see, for example, [2]). This makes sense: A binding interface adds additional constraint that reduces the rate at which mutations accumulate. However, the effect is rather weak.

The values in Fig 1B are a measure of correlation, specifically a Fisher transformed correlation coefficient. They are not evolutionary rates, so they are not reflecting faster or slower evolution, rather more or less covariation of evolutionary rates over time. We are not predicting that physically interacting interfaces evolve faster than the rest of the protein, but rather that if physical interaction drives covariation in evolutionary rates over time, their correlation would be stronger between pairs of physically interacting domains. In response, we have used clearer language in the figure caption and reorganized labels in Figure 1B to clearly show that the values are correlations. Revised Figure 1 Legend:

“Overview of experimental schema and hypotheses. Proteins that share functional/physical relationships have similar relative rates of evolution across the phylogeny, as shown in (A) with SMC5 and SMC6. The color scale along the bottom indicates the relative evolutionary rate (RER) of the specific protein for that species compared to the genome-wide average. A higher (red) RER indicates that the protein is evolving at a faster rate than the genome average for that branch. Conversely, a lower (blue) RER indicates that protein is evolving at a slower rate than the genome average. The ERC (right) is a Pearson correlation of the RERs for each shared branch of the gene pair. (B) Suppose the correlation in relative evolutionary rates between two proteins is due to compensatory coevolution and physical interactions. In that case, the correlation of their rates (ie. ERC value) would be higher for just the amino acids in the physically interacting domain. (C) Outline of experimental design. Created with Biorender.com”

All in all, I'm fine with the analysis the authors perform, and I think the conclusions make sense, but the authors have to put some serious effort into reading the relevant literature and then reassess whether they are actually asking a meaningful question and, if so, whether they're doing the best analysis they could do or whether alternative hypotheses or analyses would make more sense.[1] https://www.ncbi.nlm.nih.gov/pmc/articles/PMC4523088/[2] https://www.ncbi.nlm.nih.gov/pmc/articles/PMC4854464/
**Recommendations for the authors:**

**Reviewer #1 (Recommendations For The Authors):**
Major comments(1) Numerous parameters influence ECR calculation. The authors note that their use of a large dataset of budding yeast provides sufficient statistical power to calculate ECR. I agree with that. However, a discussion of other parameters needs to be improved, especially when comparing the present study to others like Kann et al., Hakes et al., and Jothi et al.. For example, what is the evolutionary breadth and depth used in the Kann, Hakes, Jothi and other studies? How does that compare to the present study? Budding yeast evolve rapidly with gene presence/absence polymorphisms observed in genes otherwise considered universally conserved. Is there any reason to expect different results in a younger, slower-evolving clade such as mammals? There is potential to acknowledge and discuss other parameters that may influence ECR, such as codon optimization and gene/complex "essentiality," among others.

More discussion of these parameters is a good idea. We have added the number and phylogeny of species used in the previous studies in the discussion paragraph starting with “Previous studies attributed varying degrees of evolutionary rate covariation signal to physical interactions between proteins.” We also like the idea of studying the effect of younger and more slowly evolving clades as opposed to the contrary, but currently we lack the required number of datasets to do this.

We have also added more discussion and clarification of potential non-physical forces leading to ERC correlations in the introduction.

Minor comments(1) It would be good to add a citation to the second sentence of the first paragraph, which reads, "It has been observed that some genes have rates that covary with those of other genes and that they tend to be functionally related."

Added citation to Clark et al. 2012

(2) In the last sentence of the first paragraph of the introduction, ERC is discussed in the context of only amino acid divergence, however, there is no reason that DNA sequences can't be used, especially if ERC is being calculated among species that are less ancient than, for example, Saccharomycotina yeasts. Thus, it may be more accurate to suggest that ERC measures how correlated branch-specific rates of sequence divergence are with those of another gene.

Nice suggestion to generalize. We have made this change.

(3) ERC was not calculated in reference #2. For the sentence "Protein pairs that have high ERC values (i.e., high rate covariation) are often found to participate in shared cellular functions, such as in a metabolic pathway2 or meiosis3 or being in a protein complex together," I think more appropriate citations (including inspiring work by the corresponding author) would bea) Coevolution of Interacting Fertilization Proteins (https://journals.plos.org/plosgenetics/article?id=10.1371/journal.pgen.1000570)b) Evolutionary rate covariation analysis of E-cadherin identifies Raskol as a regulator of cell adhesion and actin dynamics in *Drosophila* (https://journals.plos.org/plosgenetics/article?id=10.1371/journal.pgen.1007720)c) An orthologous gene coevolution network provides insight into eukaryotic cellular and genomic structure and function (https://www.science.org/doi/10.1126/sciadv.abn0105)d) PhyKIT: a broadly applicable UNIX shell toolkit for processing and analyzing phylogenomic data (https://academic.oup.com/bioinformatics/article/37/16/2325/6131675)

Thank you for pointing out these works. We agree that there are more appropriate citations and we have referenced your suggested b-d.

(4) The dataset of 343 yeast species also includes outgroup taxa. Therefore, indicating that 332 species are Saccharomycotina yeast and 11 are closely related outgroup taxa may be more accurate.

Thank you for the suggestion, the following sentence has been added, citing the Shen et. al 2018 paper that the dataset was derived from:

“To investigate the discrepancy between contributions to ERC signal from co-function and physical interaction, we used a dataset of 343 evolutionarily distant yeast species. 332 of the species are Saccharomycotina with 11 closely related outgroup species providing as much evolutionary divergence as humans to roundworms3”

(5) Are there statistics/figures to support the claim that "Almost all complexes and pathways had mean ERC values significantly greater than a null distribution consisting of random protein pairs"?

This is shown in supplementary figure 1. A reference to this figure was added as well as quantification within the text.

(6)Similar to the previous comment, can quantitative values be added to the statement "While protein complexes appear to have higher mean ERC scores than the pathways..."?

The median of the mean ERC scores for protein complexes is 5.366 while the median for the mean ERC score in pathways is 4.597. This quantification has been included in the text: “While protein complexes have higher mean ERC scores (median 5.366) than the pathways (median 4.597), the members of a given complex are also co-functional, making interpretation of the relative contribution of physical interactions to the average ERC score difficult”

These quantifications are were also added to the figure caption for figure 2A

(7) A semantic point: In the sentence "The lack of significance in the global permutation test shows that the...", I recommend saying that the analysis suggests, not shows, because there is potential for a type II error.

Good suggestion, we have made this change.

(8) The authors suggest that shared evolutionary pressures, "and hence shared levels of constraint," drive signatures of coevolution. The manuscript does not delve into selection measures (e.g., dN/dS). Perhaps it would be more representative to remove any implication of selection.

We have added better language to clarify that discussion of selection is purely a hypothesis and that selection is not probed in our analyses.

“Previous work finds evidence that relaxation of selective constraint can lead to drastic rate variation and hence covariation6. Rather, the greater and consistent contribution comes from non-physical interaction drivers that could include variation in essentiality, expression level, codon adaptation, and network connectivity. These non-physical forces would be under shared selective pressures and hence shared levels of constraint, the result of which was elevated ERC between non-interacting proteins, as visible in our study of genetic pathways that do not physically interact (Figure 2).”

**Reviewer #2 (Recommendations For The Authors):**
Major comments:-Title: In my opinion, the title of the manuscript is a somewhat misleading summary of the results of this paper. In the majority of the analyses in this paper, physical interactions do account for a significantly outsized portion of the ERC signature. The current title downplays the consistent (although sometimes small effect-sized) result that physically interacting domains do show higher ERC than non-physically interacting domains by every statistical measure employed in this paper to compare physical vs non-physical interactions. The authors' interpretation of their results within the manuscript body is that the effect of physical interactions is an inconsistent, weak, and non-generalizable driver of ERC. I generally agree with the authors' interpretations, but the nuance of these interpretations is lost in the title of the paper. I would suggest rewording the title to try to capture the nuance or at least be subjectively accurate. For example, stating that "...physical interactions are not the sole driving force.." is inarguably accurate based on these results.As an alternative title, I would suggest focusing on an important takeaway from the paper: ERC is a reliable predictor of co-functional interactions but not necessarily physical interactions. I agree with the statement that "there is not a strong enough signal to confidently call an interaction physical or not and would be of little value to an experimentalist wanting to infer interacting domains" and I think that a title that emphasizes this idea would be more accurate and impactful.

Great suggestion. We agree that the current title is downplaying the minutiae of the method and the signal we capture with it, we have used your suggested title.

There are an outsized number of complexes that had ROC-AUCs greater than 0.5 which is why we performed the permutation tests to determine how significant each of the individual ROC-AUCs were given the differing number of protein/domain pairs in each complex. Between the statistical methods used only 3 of the 17 complexes ranked physical interactions significantly higher than non-physically interacting domains in every analysis. Even among the 3that were statistically significant some of the physically interacting domains still fell among the bottom portion of theERC scores for that complex (Figure 5: MCM and CUL8 complexes)This is why we concluded that physical interactions are not the sole driving force of the signal captured by ERC.

-Abstract: related to my preceding comment, the word "negligible" in the abstract is misleading. If physical interactions were truly entirely negligible, the comparisons of physically interacting vs non-physically interacting domains would yield 0.5. Instead, these comparisons always yielded results greater than 0.5. Consider rewording.

Thank you for the suggestion this phrasing has been changed to “Therefore, we conclude that coevolution due to physical interaction is weak, but present in the signal captured by ERC”

We agree that “negligible” may be too strong of a word, however, the comparisons do not always yield results greater than 0.5.

5 of the 17 complexes do not reach the 0.5 threshold for the initial ROC analysis and even among those that do, only 4 had significantly high ROC-AUCs. You are correct that the signal is not completely negligible which is why we continued by determining if the physical interaction was driving high ERC only within proteins (Figure 5)

-Figure 3: I think there may be an error in the domain labeling in Figure 3. The comparison between OKP1_2 and AME1_3 is the highest ERC value in the matrix. From the complex structure, it appears that OKP1_2 and AME1_3 are two helix domains that appear to physically interact. However, in the ERC matrix, they are not shaded to indicate they are a physical interaction pair. Please double-check that the interacting domains are properly annotated, since mis-annotation would have a large impact on the interpretation of this figure with respect to the overall question the paper addresses.

Thank you for catching this - fixed.

Minor comments:-Methods: "The full ERC pipeline can be found at (Github)." Provide github URL here?Thanks for the catch, fixed-Discussion: "Evidence for physical coevolution however was tempered by a global permutation test, which did not reach significance, indicating that this inference is sensitive to approach and further underlines the relatively weak contribution of physical coevolution." The word "relatively" may not be a good choice of words. In comparison to what? As is, the phrasing could be interpreted as implying "in comparison to non-physical interactions". This would not be accurate, because the results show that in general, physical interactions are a stronger contributor to ERC (consistent trend but varied significance, depending on methodology) than non-physical interactions.

Thank you for your help with clarification. The word relatively was removed.

However, we do not agree that in general physical interactions are a stronger contributor to ERC than non-physical interactions (such as gene expression, codon adaptation, etc.). In all of our statistical tests a maximum of 5 of the 17 complexes ranked physical interactions significantly higher than non-physical interactions. While the ROC-AUC is greater than 0.5 for 12 of the 17 complexes only 4 of those were significant.

-I have not seen Fisher-transformed correlation coefficients used in the context of ERC. I understand that it's helpful in normalizing the results so that they are comparable between ERC comparisons with differing numbers of overlapping branches (i.e. points on a linear correlation plot). A reference of where the authors got this idea or a little more verbiage to describe the rationale would be helpful. On a related note, I would expect that using linear correlation p-value instead of R-squared would account for differences in overlapping branches, eliminating the need to apply fisher-transformation. It would be helpful for the authors to outline their rationale for using a correlation coefficient rather than a P-value.

We agree that this method could be made clearer. We made a methodological choice to use Fisher transformation over linear correlation p-value. Both methods should achieve the same end result by taking the number of branches into consideration. We have added additional explanation to the results section “Both protein pathways and complexes have elevated ERC”:

“ERC was calculated for all pairs of the 12,552 genes. For each pair the correlation is Fisher transformed to normalize for the number of shared branches that contribute to the correlation. This normalization is necessary to reduce false positives that have high correlation solely due to a small number of data points. This normalization also allows for direct comparison of ERC between gene pairs that have differing numbers of branches contributing to the score.”

We also added additional explanation in the methods section including the formula used to calculate the Fisher transformation

-Did the authors use Pearson or Spearman correlation coefficient?

Pearson. We clarified this in the methods section, “Calculating evolutionary rate covariation” :“Evolutionary rate covariation is calculated by correlating relative evolutionary rates (RERs) between two gene trees using a Pearson correlation.”

-Did the authors explore ERC between domains within a single protein? Do domains within a protein exhibit ERC? I would expect that they do. If they do, this could likely be attributed to linkage/genetic hitchhiking, representing a new angle/factor beyond physical interaction that could lead to ERC. This is just an idea for a future analysis, not necessarily a request within the scope of the present paper.

We did calculate the ERC between domains of a single protein but did not include them in the analysis since they didn’t address the specific question we posed. As expected they are highly correlated, and past unpublished studies in the lab do find a very weak, but detectable genome-wide, signature of rate covariation between neighboring colinear genes on a chromosome. That signal was however so weak as to be eclipsed by true functional relationships, when present.

**Reviewer #3 (Recommendations For The Authors):**
Please read the literature and revise accordingly.

We understand the confusion surrounding previous literature on the relationship between expression levels and evolutionary rates when comparing between different proteins. Those studies clearly showed how expression level is highly predictive of a given protein’s average evolutionary rate. However, we are studying the change in evolutionary rate over branches for single proteins. This is inherently different because we’re following rate fluctuations in the same protein over time. To our knowledge it has not yet been shown that expression level commonly varies enough over time to produce large rate variations over time in the same protein, and if it is responsible for the correlations of rate we observe between co-functional proteins. It is however reasonable to expect that what governs between-protein differences in rate could also contribute to between-branch differences (over time for a single protein). In fact, our earlier study approached this (Clark et al. Genome Research 2012). We expect expression level could influence rate over time and lump its effect together with general non-physical forces, such as selection pressures. We recognize we could do better in defining more of the non-physical forces and the past literature. We added the following section to the introduction and many other clarifying statements throughout the manuscript:

“For the purposes of this study, the forces that contribute to correlated evolutionary rates are grouped into two bins, physical and non-physical. The physical force is coevolution occurring at physical interaction interfaces. Non-physical forces include gene co-expression, codon adaptation, selective pressures, and gene essentiality. There is a well accepted negative relationship between gene expression and rate of protein evolution where genes that are highly expressed generally have slower rates of evolution14,15. However, Cope et al.16 found that there is a weak relationship between both gene expression and the number of interactions a protein has with the coevolution of expression level. Conversely, they found a strong relationship between proteins that physically interact and the coevolution of gene expression. These findings illuminate the difference between the strong relationship of gene expression level on the average evolutionary rate of a protein and the weak contribution of gene expression level to correlated evolutionary rates of proteins across branches. The finding that physically interacting proteins have strong expression level coevolution brings to question how much coevolution of physically interacting proteins contributes to overall covariation in protein evolutionary rates.”